# Black Ginseng Extract Exerts Potentially Anti-Asthmatic Activity by Inhibiting the Protein Kinase Cθ-Mediated IL-4/STAT6 Signaling Pathway

**DOI:** 10.3390/ijms241511970

**Published:** 2023-07-26

**Authors:** Yu Na Song, Jae-Won Lee, Hyung Won Ryu, Jae Kyoung Lee, Eun Sol Oh, Doo-Young Kim, Hyunju Ro, Dahye Yoon, Ji-Yoon Park, Sung-Tae Hong, Mun-Ock Kim, Su Ui Lee, Dae Young Lee

**Affiliations:** 1Natural Product Research Center, Korea Research Institute of Bioscience and Biotechnology (KRIBB), Cheongju 28116, Republic of Korea; alsrud5354@kribb.re.kr (Y.N.S.); suc369@kribb.re.kr (J.-W.L.); ryuhw@kribb.re.kr (H.W.R.); zkx2@kribb.re.kr (E.S.O.); rose73@kribb.re.kr (D.-Y.K.); parkgyun98@kribb.re.kr (J.-Y.P.); mokim@kribb.re.kr (M.-O.K.); 2Department of Biological Sciences, College of Bioscience and Biotechnology, Chungnam National University, Daejeon 34134, Republic of Korea; rohyunju@cnu.ac.kr; 3Rpbio Research Institute, Rpbio Co., Ltd., Suwon 16229, Republic of Korea; ljk1200@rpcorp.co.kr; 4Department of Herbal Crop Research, National Institute of Horticultural and Herbal Science, RDA, Eumseong 27709, Republic of Korea; dahyeyoon@korea.kr; 5Department of Anatomy & Cell Biology, Department of Medical Science, College of Medicine, Chungnam National University, Daejeon 35015, Republic of Korea; mogwai@cnu.ac.kr

**Keywords:** asthma, black ginseng extract, PKCθ, STAT6, Th2 cytokines

## Abstract

Asthma is a chronic inflammatory lung disease that causes respiratory difficulties. Black ginseng extract (BGE) has preventative effects on respiratory inflammatory diseases such as asthma. However, the pharmacological mechanisms behind the anti-asthmatic activity of BGE remain unknown. To investigate the anti-asthmatic mechanism of BGE, phorbol 12-myristate 13-acetate plus ionomycin (PMA/Iono)-stimulated mouse EL4 cells and ovalbumin (OVA)-induced mice with allergic airway inflammation were used. Immune cells (eosinophils/macrophages), interleukin (IL)-4, -5, -13, and serum immunoglobulin E (IgE) levels were measured using an enzyme-linked immunosorbent assay. Inflammatory cell recruitment and mucus secretion in the lung tissue were estimated. Protein expression was analyzed via Western blotting, including that of inducible nitric oxide synthase (iNOS) and the activation of protein kinase C theta (PKCθ) and its downstream signaling molecules. BGE decreased T helper (Th)2 cytokines, serum IgE, mucus secretion, and iNOS expression in mice with allergic airway inflammation, thereby providing a protective effect. Moreover, BGE and its major ginsenosides inhibited the production of Th2 cytokines in PMA/Iono-stimulated EL4 cells. In EL4 cells, these outcomes were accompanied by the inactivation of PKCθ and its downstream transcription factors, such as nuclear factor of activated T cells (NFAT), nuclear factor kappa B (NF-κB), activator of transcription 6 (STAT6), and GATA binding protein 3 (GATA3), which are involved in allergic airway inflammation. BGE also inhibited the activation of PKCθ and the abovementioned transcriptional factors in the lung tissue of mice with allergic airway inflammation. These results highlight the potential of BGE as a useful therapeutic and preventative agent for allergic airway inflammatory diseases such as allergic asthma.

## 1. Introduction

Asthma is an inflammatory disease caused by genetic or environmental risk factors. Poorly controlled or untreated asthma can lead to respiratory failure and death. Untreated asthma can also cause chronic inflammation of the lungs and airway structural remodeling. Currently, there are more than 339 million people suffering from asthma worldwide [1]. Although inhaled corticosteroids (ICSs) alone or in combination with bronchodilators (Beta-2 agonists and anticholinergics) have yielded positive results in the treatment of asthma, long-term and high-dose inhaled corticosteroid use can potentially lead to side effects such as diabetes, fractures, and pneumonia [2]. More than half of patients with severe asthma cannot control their disease effectively and remain at risk of exacerbations [3]. The failure of conventional therapies has led to the development of safer, more effective drugs for long-term asthma management. Therefore, complementary and alternative medicines, such as herbal remedies and biologicals, are being developed and are widely used for the treatment of asthma worldwide [4,5]. Some herbal remedies that are safe, effective, and comparable to standard medications, have shown significant improvements in clinical symptom scores [6].

Ginseng (*Panax ginseng* Meyer) is widely used as a traditional herbal medicine in East Asia, including Korea [7]. The main bioactive ingredients of ginseng, namely, ginsenosides, are valuable therapeutic agents for lung inflammatory diseases, including asthma [8,9]. Fresh Korean ginseng is processed over three times of steaming/drying, resulting in a color change to black ginseng (BG) [10]. This repetitive heating process increases the content of nonpolar ginsenosides, which contributes to the enhancement of various physiological activities [11]. Although our previous studies reported that black ginseng extract (BGE) protects against inflammation and damage in the lungs of experimental disease mice [12,13], the preventive efficacy and mechanism of action of BGE in a model of ovalbumin (OVA)-induced lung inflammation remain unknown.

T helper (Th) 2 cells play a key role in the development and progression of allergic asthma [14]. Th2 cell-derived cytokines, such as interleukin (IL)-4, -5, and -13, cause airway inflammation, mucus hypersecretion, and immunoglobulin E (IgE) synthesis in response to allergens [14]. The secretion of Th2 cytokines is markedly increased in the bronchoalveolar lavage fluid (BALF) in both OVA-induced mice with allergic airway inflammation and patients with asthma [1,15]. Modulating Th2 cell immunity is recognized as a promising therapeutic target in the treatment of asthma, and several drugs using monoclonal antibodies against Th2 cytokines have already been approved [16]. In addition, other similar therapeutic approaches are currently being investigated [14,15].

Activation of the IL-4/signal transducer and activator of transcription 6 (STAT6) axis promotes the expression of Th2-specific cytokines such as IL-4, -5, and -13 [17]. Following activation of the T-cell receptor, the Ca^2+^-dependent transcription factor NFAT is activated via dephosphorylation and moves to the nucleus, where it induces the expression of IL-4 [18,19]. Subsequently, autocrine IL-4 induces the transcriptional activity of STAT6 and GATA binding protein 3 (GATA3), which are essential for the expression of Th2 cytokines [20]. Therefore, therapeutic agents that negatively modulate the activity of transcription factors that regulate the IL-4/STAT6 axis, and their upstream signaling kinases, could be used as effective approaches to treat Th2 cell-mediated inflammation in asthmatic patients.

Protein kinase C (PKC) is a member of the serine/threonine kinase family and is closely related to lung inflammatory diseases. The PKC family comprises 15 isozymes in humans, which are classified into three categories: classical (α, β1, βII, and γ), novel (δ, ε, θ, and η), and atypical (ζ, λ, ι, and μ) [21]. PKCθ belongs to the calcium-independent novel PKC subfamily. PKCθ is exclusively expressed in T lymphocytes, skeletal muscle cells, and platelets. Particularly, PKCθ is highly expressed in T cells, involved in T-cell activation and proliferation, and modulates T-cell activation in several inflammatory disease models, such as multiple sclerosis, inflammatory bowel disease, and allergic asthma. Asthma-related transcription factors that are activated by PKCθ include STAT6, NFAT, GATA3, and nuclear factor kappa B (NF-κB) [22,23]. PKCθ-deficient mice exhibit a blockage in allergen-induced Th2 cell responses [24] and impaired IL-4 expression [23]. Moreover, pharmacologic inhibition of PKCθ by a specific inhibitor (Compound 20, C20) significantly reduces allergic lung inflammation [22]. Thus, inhibition of PKCθ activity and its downstream transcription factors may be promising therapeutic targets for asthma treatment.

As shown in Figure 1, which is drawn based on the signaling molecules mentioned above, this study aimed to evaluate the anti-asthmatic activity of BGE and elucidate its molecular mechanisms using OVA-induced mice with allergic airway inflammation and a murine T-cell line, namely, EL4. Additionally, the effects of the three major ginsenosides (Rg3, Rg5, and Rk1) of BGE on PKCθ and IL-4/STAT6 signal activation were determined.

## 2. Results

### 2.1. BGE Decreases Inflammatory Cells and Mediators in OVA-Sensitized Mice with Allergic Airway Inflammation

To analyze the contents of the standardized BGE used in this study, we performed high-performance liquid chromatography (HPLC) profiling, which revealed that standardized BGE mainly contains the ginsenosides Rg3, Rg5, and Rk1 (Appendix A).

It is well known that OVA-stimulated mice with allergic airway inflammation increase the recruitment of immune cells such as eosinophils and macrophages in the BALF of lung tissue [25]. Thus, we investigated the effect of BGE administration on the number of inflammatory cells. In these results, the OVA-induced asthma model exhibited a marked increase in the number of eosinophils and macrophages, compared with the normal control (NC) mice (Figure 2A,B, second bar, respectively). However, BGE and dexamethasone (DEX) administration significantly reduced the number of eosinophils and macrophages compared to OVA-induced mice with allergic airway inflammation. BGE was comparable to that of a 1 mg/kg administration of DEX, which was used as a positive control (Figure 2A,B).

Since elevated Th2 cytokine, IgE, and iNOS expressions are common characteristics of airway inflammation in mice with allergic airway inflammation and patients with asthma [26,27], we investigated whether BGE administration inhibited Th2 cytokine, IgE, and iNOS expression. Notably, a marked increase in cytokine expression levels (IL-4, -5, and -13) in BALF and IgE levels in serum were observed in OVA-sensitized mice with allergic airway inflammation compared with mice in the NC group (Figure 2C–F, second bar, respectively). However, BGE or DEX treatment significantly suppressed the expression levels of cytokines and serum IgE. Moreover, BGE and DEX inhibited OVA-increased iNOS expression in the lung tissue lysates (Figure 2G). These results showed that BGE possesses potentially anti-asthmatic activity by suppressing the recruitment of immune cells (eosinophils/macrophages), Th2 cytokine secretion, serum IgE production, and iNOS expression levels in the OVA-sensitized asthma mouse model.

### 2.2. BGE Decreases Inflammatory Cell Recruitment and Mucus Secretion in Lung Tissues of OVA-Sensitized Mice with Allergic Airway Inflammation

OVA inhalation can induce Th2 cell-mediated allergic inflammation, that is, influx of inflammatory cells and mucus hypersecretion in the airways in mice [28]. Thus, we investigated the potency of BGE administration on the recruitment of inflammatory cells and mucus production using hematoxylin and eosin (H&E) and periodic acid–Schiff (PAS) staining, respectively. The results of H&E staining clearly showed increased bronchial wall thickening and immune cell infiltration in OVA-sensitized mice with allergic airway inflammation, in contrast to the NC group. However, BGE and DEX treatment alleviated these histopathological changes (Figure 3A). Furthermore, PAS staining showed increased mucus production in the lung epithelium of the OVA-sensitized group compared with that in the NC group (Figure 3B). Treatment with BGE or DEX appeared to inhibit airway mucus production by suppressing the increase in PAS-positive epithelial areas. These findings indicate that BGE ameliorates respiratory inflammatory responses by inhibiting the accumulation of airway inflammatory cells and mucus secretion in OVA-sensitized mice with allergic airway inflammation.

### 2.3. BGE Suppresses the Expression of Th2 Cytokines in EL4 Cells

To study the T-cell activation-related cytokine expression changes after BGE treatment, we used a model system in which mouse EL4 cells were stimulated with PMA and Iono (PMA/Iono). The EL4 cells were treated with BGE at concentrations of 20, 40, and 80 μg/mL for 24 h, and cell viability was tested using a CCK-8 assay. No significant decrease in cell viability was observed after BGE treatment up to a concentration of 80 μg/mL (Figure 4A). Aberrant Th2-mediated asthma is associated with Th2 cytokines such as IL-4, -5, and -13. Expectedly, PMA/Iono potently induced IL-4, -5, and -13 expression in EL4 cells, whereas BGE dose-dependently inhibited the secretion of IL-4, -5, and -13, as measured by an enzyme-linked immunosorbent assay (ELISA) (Figure 4B–D). To determine whether BGE could regulate the expression of these cytokines at the transcriptional level, we investigated the effect of BGE on *Il-4*, *-5*, and *-13* mRNA expression using qRT-PCR in PMA/Iono-stimulated EL4 cells. Consistent with the ELISA results, BGE significantly inhibited the mRNA expression of PMA/Iono-induced *Il-4, -5,* and *-13* in a concentration-dependent manner (Figure 4E–G). These results demonstrated that BGE could regulate the expression of Th2 cytokines by suppressing both transcriptional and translational expression levels in PMA/Iono-stimulated EL4 cells.

### 2.4. BGE Inhibits the Activation of PKCθ and Inflammatory Transcription Factors (NFAT, NF-κB, STAT6, and GATA3) In Vitro and In Vivo

The gene expression of Th2 cytokines, such as IL-4, -5, and -13, is induced by the activation of several transcription factors, such as NFAT, GATA3, and STAT6 [29,30]. Thus, we investigated whether BGE affects signaling molecules in PMA/Iono-stimulated EL4 cells. Western blotting was performed using antibodies against NFAT, GATA3, p-STAT6, and STAT6. When PMA/Iono was added to EL4 cells for 24 h, an increase in NFAT and GATA3 was observed in the nucleus, whereas pretreatment with BGE significantly reduced their nuclear translocation (Figure 5A). Moreover, BGE potently suppressed PMA/Iono-induced STAT6 phosphorylation without altering the expression of the total STAT6 protein (Figure 5B). These results showed that BGE suppresses the activation of asthma-related transcription factors, such as NFAT, GATA3, and STAT6, suggesting a molecular mechanism underlying the downregulation of Th2 cytokine expression.

Since PKCθ is necessary for efficient activation of transcription factors such as NFAT, NF-κB, GATA3, and STAT6 during allergic asthma reactions [22,23,31], we investigated whether BGE or C20 (a specific PKCθ inhibitor) could inhibit the activation of PKCθ and asthma-associated transcription factors in PMA/Iono-stimulated EL4 cells. Western blot analysis was performed using targeted antibodies for p-PKCθ, PKCθ, NFAT, p-NF-κB, p-STAT6, STAT6, and GATA3. As shown in Figure 5C, PMA/Iono exposure for 30 min increased activation of PKCθ, NFAT, and NF-κB; however, BGE or C20 considerably inhibited these activities in a dose-dependent manner without changing total PKCθ expression (Figure 5C). Moreover, BGE or C20 inhibited STAT6 phosphorylation and GATA3 expression without altering the total STAT6 expression in EL4 cells treated with PMA/Iono for 24 h (Figure 5D). In the lung tissues of OVA-induced allergic airway inflammation, PKCθ, STAT6, and NF-κB phosphorylation and GATA3 and NFAT expression levels were increased but were effectively inhibited by BGE without changing the total PKCθ and STAT6 expression (Figure 5E). The positive control, DEX, also showed a similar inhibitory effect. Overall, our results found that BGE inhibits allergic airway inflammation by negatively regulating the activation of PKCθ and asthma-related transcription factors (NFAT, NF-κB, GATA3, and STAT6) in mice with allergic airway inflammation as well as in EL4 cells.

### 2.5. Ginsenosides (Rg3, Rg5, and Rk1) Isolated from BGE Inhibit PKCθ and Its Downstream IL-4/STAT6 Signaling Pathway in EL4 Cells

We investigated whether each of the three major ginsenosides (Rg3, Rg5, and Rk1) isolated from BGE had a suppressive effect on IL-4 expression via the phosphorylation of signaling molecules such as PKCθ and STAT6. A specific PKCθ inhibitor, C20, was also included in the experiment. The results of the cell viability assay showed that none of the compounds had any cytotoxicity at the corresponding concentrations in EL4 cells (Figure 6A). Pretreatment with each of the three ginsenosides and C20 significantly inhibited PKCθ (Thr538) phosphorylation (Figure 6B). Expectedly, the three ginsenosides and C20 inhibited PMA/Iono-induced IL-4 expression (Figure 6C) and decreased STAT6 phosphorylation without affecting total STAT6 expression (Figure 6D). These results demonstrate that the three ginsenosides inhibited IL-4/STAT6 signal activation by suppressing the phosphorylation of PKCθ, similar to the mode of action of C20, in PMA/Iono-induced EL4 cells.

## 3. Discussion

Until now, ICSs have been consistently used as an effective drug for the treatment of asthma. However, long-term or high doses of ICSs have some limitations due to side effects such as osteoporosis and pneumonia [32]. Thus, there is an increasing recognition of the importance of remedial plants to diminish asthma symptoms in a more effective and safe manner [33]. The processed ginseng has been considered an active alternative or complementary to standard therapies for lung inflammatory diseases such as lung cancer [34], asthma [35], and chronic obstructive pulmonary disease [13]. However, the potential therapeutic mechanism of BGE in allergic asthma-related lung inflammation is poorly understood.

In this report, for the first time, we describe that standardized BGE and its major ginsenosides (Rg3, Rg5, and Rk1) possess effective anti-asthmatic activity by showing a relationship between PKCθ and asthma-associated transcription factors in PMA/ionomycin-stimulated EL4 cells or OVA-exposed mice with allergic airway inflammation.

Most asthmatic patients and mice with allergic airway inflammation express high levels of Th2 cytokines, which affect IgE production and iNOS expression in inflamed lungs [1,36,37]. The cytokine IL-4 promotes the synthesis and release of allergen-specific IgE from B cells, and IL-5 induces eosinophilic inflammation. Airway hyperresponsiveness (AHR) exacerbation, immune cell influx, and mucus secretion in asthma are all controlled by IL-13 [38]. Moreover, IL-4 and -13 promote the production of nitric oxide in inflammatory cells by increasing the transcriptional expression of iNOS [39]. In patients with asthma, the level of allergen-specific IgE production is a pathological marker correlated with the severity of the disease [16], and iNOS expression is involved in the inflammation of the upper and lower airways [39]. Therefore, modulating allergic hallmarks, such as IgE, iNOS, and Th2 cytokines, is an important measure for the amelioration of allergic asthma in the lungs of OVA-induced mice. In this study, we showed that BGE significantly suppressed Th2 cytokine levels, serum IgE production, and iNOS expression, which is similar to the effects of DEX in OVA-induced asthma mice. Moreover, BGE effectively reduced the infiltration of immune cells and mucus secretion in the lung tissue of OVA-induced mice with allergic airway inflammation. Consistent with in vivo results, BGE decreased the secretion of Th2 cytokines in PMA/Iono-stimulated EL4 cells. These results suggest that BGE exhibits protective activity against allergic airway inflammation induced by allergens.

Of the PKC isozymes, PKCζ has been reported to be crucial for the biology of T cells and the development of allergic disorders. Some natural compounds can affect PKCζ through epigenetic mechanisms [40,41,42,43]. Moreover, inhibition of PKCδ activation was found to reduce asthma attacks in a murine model [44]. PKCθ activity in particular is important for Th2-mediated responses in allergic lung inflammation [45]. PKCθ-deficient mice failed to induce allergic responses, such as Th2 secretion, IgE production, and AHR in an atopic asthmatic model [24]. Moreover, the administration of C20 reduced Th2 cell-mediated allergic inflammation in house dust mite-induced allergic asthma [22]. The PKCθ enzyme is required for the activation of inflammatory transcription factors, such as NFAT and NF-κB, which induce IL-4 and iNOS expression [22,23,46]. Indeed, PKCθ–deficient T cells indicated inactivation of NF-κB and NFAT, which are important for the production of the signature cytokine IL-4 in Th2 cells [47,48,49]. Moreover, PKCθ-deficient mice or C20 administration showed a decrease in IL-4 in allergic lung inflammation by reducing NFAT expression levels in vivo [22]. Increased levels of IL-4 initiate STAT6 activation (IL-4/STAT6), which induces the activation of GATA3, a master regulator driving Th2 cytokine expression [20]. Therefore, PKCθ and its downstream transcription factors are accountable for the control of asthma-related inflammatory responses and tissue injury [22], suggesting that standardized extracts or natural bioactive compounds targeting these signaling molecules may modulate symptoms of pulmonary inflammatory disease. The results of this study showed that BGE exerts anti-inflammatory activities by suppressing PKCθ and asthma-associated transcription factors, similar to C20, in PMA-stimulated EL4 cells [22,23]. Notably, C20 also inhibits the activation of STAT6 and GATA3, which promote the expression of Th2 cytokines, supporting the idea that they are downstream molecules of PKCθ. Consistent with in vitro results, BGE suppressed the activation of PKCθ and inflammatory transcription factors in lung tissues of OVA-treated mice with allergic airway inflammation. In addition, the major three ginsenosides contained in BGE inhibited PKCθ phosphorylation similarly to C20, thereby reducing the activation of STAT6, which induces IL-4 expression. These results indicate that ginsenosides decrease the expression of Th2 cytokines by suppressing the activation of PKCθ and its downstream transcription factors.

Like most studies, the present study has some limitations. In this study, we only used EL4 cells, which are a mouse T-cell lymphoma cell line, and additional cell lines such as primary human cells were not used. Although EL4 cells are widely used as a model system for studying the expression of T-cell markers and molecular mechanisms by activating the T-cell receptor using substances such as PMA and Iono [50,51], primary human cells from patients with asthma will provide more valuable supporting evidence for the anti-inflammatory effect of BGE. Despite these limitations, our study provides a significant contribution to the existing literature. However, future studies should be designed to closely relate to human application tests by using primary human cells from patients with asthma.

In summary, we found that BGE exerts anti-inflammatory activity both in vitro and in vivo by reducing the activation of PKCθ and asthma-associated transcription factors (NFAT, NF-κB, STAT6, and GATA3). Since BGE also effectively suppressed the allergic inflammatory response induced by OVA, it may be used as a supplement for patients who are at risk for the development of allergic diseases caused by allergens.

Therefore, we propose that BGE could be a promising therapeutic agent for allergic inflammation.

## 4. Materials and Methods

### 4.1. Preparation of BGE

The ginseng roots were processed using a method described in the Korean Pharmacopoeia (KP). The raw material of *P. ginseng* was deposited as a voucher specimen (RDA21-01) in the herbarium of the Department of Herbal Crop Research, National Institute of Horticultural and Herbal Science (NIHHS), RDA, Republic of Korea [13]. Black ginseng was made by repeating the process of steaming dried ginseng roots at 95 ± 2 ℃ for 6 h and then drying them at 40 °C for 8 h four times. BGE powder was prepared by spray drying after extraction at 85 °C for 8 h with 30%, 40%, or 60% ethanol and 100% water on dried black ginseng. The sum of ginsenosides Rg3, Rk1, and Rg5 was 7.36 mg/g, which is suitable for BGE’s quality control criteria, and a representative HPLC chromatogram of BGE is shown in Appendix A.

### 4.2. Analysis of Chromatographic Condition for BGE

Experiments on BGE were performed on a Waters Arc high-performance liquid chromatography system (HPLC; Waters, Milford, MA, USA), including a quaternary solvent manager and a UV/Visible Detector (UVD). Gradient elution was performed using ultrapure water (A) and acetonitrile (B). The linear gradient elution conditions were as follows: 0.0 min, 18% B; 0.0–10.0 min, 18–20% B; 10.0–30.0 min, 20–27% B; 30.0–40.0 min, 27–30% B; 40.0–55.0 min, 30–51% B; 55.0–66.0 min, 51–70% B; 66.0–71.0 min, 70–95% B; 71.0–76.0 min, 95–95% B; 76.0–77.0 min, 95–18% B; and 77.0–85.0 min, 18% B. For chemical fingerprint analysis, the chromatogram was measured while comparing the standard product and BGE under the detector wavelength (203 nm) (Appendix A).

### 4.3. Chemicals and Reagents

Phorbol 12-myristate 13-acetate (PMA; #P1585), ionomycin (Iono; #I0634), and OVA (#A5503) were purchased from Sigma-Aldrich (St. Louis, MO, USA). Aluminum hydroxide (alum, #77161) was purchased from Thermo Fisher Scientific (Waltham, MA, USA) and C20 (#S6577) from Selleckchem (Houston, TX, USA). Antibodies against phospho(p)-PKCθ (#9377), p-STAT6 (#5654), STAT6 (#5397), p-NF-κB (#3033), and NFAT (#4389) were obtained from Cell Signaling Technology (Danvers, MA, USA), and anti-PKCθ (#sc-212), anti-β-actin (#sc-47778), and anti-lamin B1 (#sc-374015) antibodies from Santa Cruz Biotechnology (Dallas, TX, USA). Anti-GATA3 (#ab61052). Anti-inducible nitric oxide synthase (iNOS; #ab136918) antibodies were purchased from Abcam (Cambridge, UK).

### 4.4. Animal Experiments

Six-week-old female BALB/c mice (weighing approximately 18 g) were purchased from Koatech Co. (Pyeongtaek, Republic of Korea) and acclimated to the environment for one week before the experiment. The mice were randomly classified into the following six treatment groups (n = 6 per group): (i) normal control (NC); (ii) OVA (OVA and alum-sensitized mice); (iii) DEX (OVA + 1 mg/kg dexamethasone); (iv) BGE 50 (OVA + 50 mg/kg BGE); (v) BGE 100 (OVA + 100 mg/kg BGE); and vi) BGE 200 (OVA + 200 mg/kg BGE). To induce airway inflammation to mimic allergic asthma, the mice were exposed to OVA sensitization and inhalation, as previously described [52]. Briefly, mice were administered a mixture of OVA and alum (day 0: 30 μg OVA and 3 mg alum; day 7: 60 μg OVA and 3 mg alum). On days 11–13, mice were administered OVA through inhalation daily for 1 h. On days 9–13, mice were orally administered BGE or dexamethasone (DEX); treatment with 1 mg/kg of DEX is widely used as a positive control for asthma treatment [53,54,55].

### 4.5. Analysis of Bronchoalveolar Lavage Fluid (BALF) of an Animal Model

To analyze the production of Th2 cytokines and serum IgE, mice were anesthetized with a mixture of Zoletil (30 mg/kg i.p.; Virbac Korea, Co, Ltd., Seoul, Republic of Korea) and xylazine (5 mg/kg i.p.; Bayer Korea, Ltd., Seoul, Republic of Korea) by intraperitoneal injection. On day 15, BALF and serum were collected using a previously reported method [53]. The levels of Th2 cytokines in BALF (R&D systems, Minneapolis, MN, USA) and IgE in serum (Biolegend, San Diego, CA, USA, # 439807) were measured by enzyme-linked immunosorbent assay (ELISA) kits, following the manufacturer’s instructions. To measure the number of inflammatory cells in BAL fluid samples, the samples were centrifuged using a CytoSpin 3 cytocentrifuge (Thermo Fisher Scientific, Waltham, MA, USA). Subsequently, the collected preparations were stained on a slide with Diff-Quik^®^ reagent (SYSMEX, Kobe, Japan, #38721), and the inflammatory cells were counted using a light microscope at 400× magnification. The number of inflammatory cells was calculated as the average of counted cells in five different fields [13].

### 4.6. Western Blot Analysis

Mice lung tissues were lysed by homogenization in RIPA lysis buffer (1/10 *w/v,* Sigma-Aldrich, St. Louis, MO, USA, #C3228) containing a protease inhibitor cocktail (Roche Ltd, Basel, Swiss, #4906837001&11836153001), as previously described [56]. In addition, EL4 cells (1 × 10^6^ cells/well) were seeded in 6-well plates, pretreated with the indicated concentrations of BGE, Rg3, Rg5, or Rk1 in serum-free medium for 2 h, and then treated with PMA (50 ng/mL) and Iono (100 ng/mL) for 30 min or 24 h. The cells were homogenized in CETi lysis buffer (TransLab, Daejeon, Republic of Korea, #TLP-121CETi) under cold conditions. At least 15 μg of whole cell lysate or 9 μg of the nuclear fraction was separated using sodium dodecyl-sulfate polyacrylamide gel electrophoresis, and Western blot analysis was carried out as previously described [57]. Blots were visualized using an Amersham Imager 680 luminescent image analyzer (GC Healthcare, Chicago, IL, USA) and quantified using densitometry (Multi Gauge software version 3.0, Fujifilm, Tokyo, Japan).

### 4.7. Histological Investigation

Mice lung tissues were collected on day 15 for analysis of histological changes. They were fixed with 10% formalin and embedded in paraffin. Subsequently, they were cut into 4 μm thick sections using a rotary microtome. The sections were then stained with hematoxylin and eosin (H&E; Sigma-Aldrich St. Louis, MO, USA) or a periodic acid–Schiff stain (PAS; IMEB, San Marcos, CA, USA). Finally, the lung tissue sections were visualized under a light microscope (H&E staining: ×100 magnification; scale bar, 100 μm; PAS staining: ×200 magnification; scale bar, 50 μm).

### 4.8. Cell Maintenance

The EL4 cell line was acquired from the American Type Culture Collection (TIB-39; Ma nassas, VA, USA) and used for less than 10 passages in all experiments. The cells were grown in Dulbecco’s Modified Eagle’s Medium (DMEM, Welgene, Gyeongsan, Republic of Korea, #LM001-05), added with 10% heat-inactivated horse serum (Thermo Fisher Scientific, Waltham, MA, USA, #26050088).

### 4.9. Cell Viability Assay

The cells were seeded in 96-well plates at a density of 5 × 10^4^ cells/well in serum-free DMEM. The cells were pretreated with the corresponding concentrations of BGE, Rg3, Rg5, and Rk1 for 2 h, and further exposed to PMA (50 ng/mL) and Iono (100 ng/mL) for 24 h. Cell viability was measured in triplicate using a Cell Counting Kit-8 (CCK-8; Dojindo Molecular Technologies, Rockville, MD, USA). Absorbance was determined using an Epoch microplate reader (BioTek Instruments, Winooski, VT, USA, #CK04) and calculated as a relative percentage (%) of the control value.

### 4.10. Determination of Secreted Cytokines in EL4 Cells

Cells (5 × 10^4^ cells/well) were seeded in 96-well plates with serum-free DMEM. The cells were pretreated with BGE or ginsenosides (Rg3, Rg5, and Rk1) for 2 h, followed by stimulation with PMA (50 ng/mL) and Iono (100 ng/mL) for 24 h. The amounts of IL-4 (#555232), -5 (#555236), and -13 (#1300CB) in the cell supernatants were evaluated by commercial ELISA kits (IL-4 and -5; BD Pharmingen, San Diego, CA, USA; IL-13; R&D Systems, Minneapolis, MN, USA), according to the manufacturer’s instructions. Absorbance was measured at 450 nm using an Epoch microplate reader (BioTek Instruments, Winooski, VT, USA).

### 4.11. Quantitative Real-Time Polymerase Chain Reaction Analysis of mRNA Expression Levels

Total RNA was separated from EL4 cells using the TRIzol reagent (Invitrogen, Waltham, MA, USA, #15596026), following the manufacturer’s protocol. Reverse transcription was performed using 2 μg total RNA, 20 pmol oligo-dT primers, and a reverse transcriptase system (Omniscript, Qiagen, Hilden, Germany, #205113). Quantitative real-time polymerase chain reaction (qRT-PCR) was performed using iQ SYBR Green supermix (#1708880) and a S1000 Thermal Cycler Real-Time PCR System (Bio-Rad, Hercules, CA, USA) in the presence of cDNA (1:25 dilution) and primers (20 pmol). The primer sequences of mouse Il-4, -5, and -13 and gapdh were as follows: mouse Il-4, 5′-ATCATCGGCATTTTGAACGAGGTC-3′ and 5′-ACCTTGGAAGCCCTACAGACGA-3′; mouse Il-5, 5′-GATGAGGCTTCCTGTCCCTACT-3′ and 5′-TGACAGGTTTTGGAATAGCATTTCC-3′; mouse Il-13, 5′-GCAACATCAACAGGACCAGA-3′ and 5′-GTCAGGTCCAGGGCTAC-3′; and mouse Gapdh, 5′-CATGTACGTTGCTATCCAGG-3′ and 5′-CTCCTTAATGTCACGCACGA-3′. PCR conditions were the same as those described previously [57]. The experiments were performed in triplicate, and the data were analyzed using the 2^−ΔΔCT^ method.

### 4.12. ELISA-Based Measurement of PKCθ Phosphorylation

PKCθ phosphorylation was measured using a PKCθ (phospho-Thr538) cell-based ELISA kit (Cat# A102530, Assay Bio-Tech, Fremont, CA, USA, #A102530), according to the manufacturer’s instructions. Briefly, EL4 cells (2 × 10^4^ cells/well) were seeded in 96-well plates and treated with BGE or ginsenosides for 2 h, followed by PMA (50 ng/mL) and Iono (100 ng/mL) for 30 min. The cells were fixed with 8% paraformaldehyde for 20 min at room temperature (RT; approximately 21–23 °C). After the removal of paraformaldehyde, quenching buffer was added, and the mixture was incubated for 20 min at RT. The cells were incubated with blocking buffer for 1 h at RT, then with primary antibodies for at least 16 h at 4 °C. After washing, the cells were treated with diluted secondary antibodies for 1 h at RT. Finally, the substrate was added to each well and incubated for 30 min at RT. After the addition of the stop solution, the optical density was measured at 450 nm using a microplate spectrophotometer (Epoch^TM^, BioTek Instruments, Winooski, VT, USA).

### 4.13. Statistical Analysis

Data are presented as the mean ± standard deviation. Statistical significance was analyzed using a two-tailed Student’s *t*-test (Microsoft Excel, Redmond, WA, USA) for in vitro experiments. Statistical significance was set at * *p* < 0.05, ** *p* < 0.01, and *** *p* < 0.001. One-way analysis of variance, followed by Tukey’s multiple comparison test, was used to analyze data from the in vivo experiments using SPSS (version 20.0; IBM Corp., Armonk, NY, USA). Single asterisks (*) represent statistical significance at *p* < 0.05.

## 5. Conclusions

Here, we demonstrate that BGE can alleviate allergic airway inflammation in OVA-induced mice with allergic airway inflammation and inactivate PKCθ and the inflammatory transcription factors associated with Th2 cytokine expression. The key findings of this study are that BGE inhibits PKCθ-mediated signaling molecules related to a central pathway in allergic airway asthma, and the corticosteroid drug DEX, used as a positive control, has similar efficacy and mechanisms to BGE. Considering the side effects of current DEX therapies, BGE could be a potential adjuvant to a safer and more effective treatment for allergic airway inflammation. Thus, we propose BGE as a promising therapeutic, pharmacological, and nutraceutical candidate for allergic asthma management. An important future direction includes validating our findings in human studies with the overall aim of evaluating the therapeutic potential of BGE in allergic asthma.

## Figures and Tables

**Figure 1 ijms-24-11970-f001:**
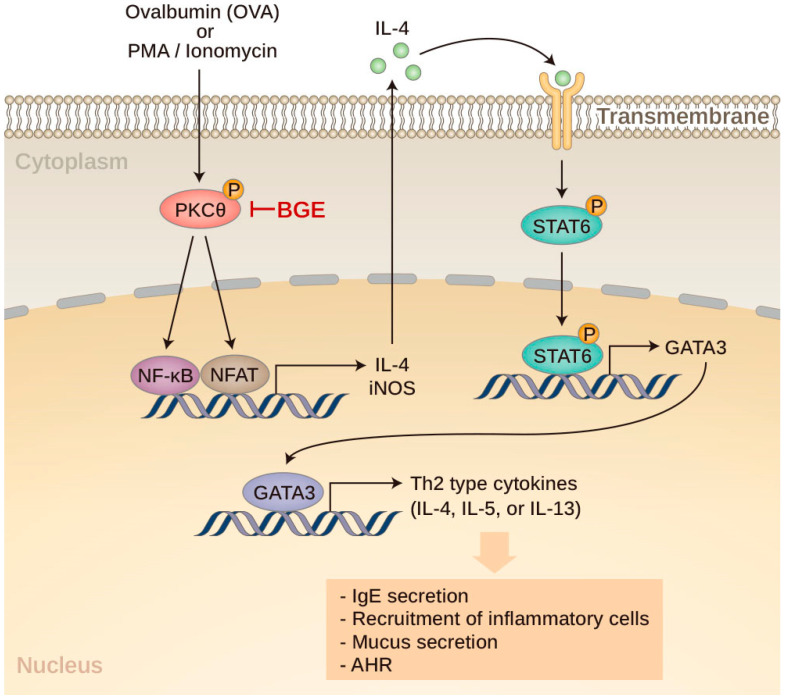
Proposed anti-inflammatory mechanism of black ginseng extract (BGE) in both phorbol 12-myristate 13-acetate plus ionomycin-stimulated EL4 cells and ovalbumin-sensitized mice with allergic airway inflammation. PMA, phorbol 12-myristate 13-acetate; IL, interleukin; PKCθ, protein kinase C theta; BGE, black ginseng extract; NF-κB, nuclear factor kappa B; NFAT, nuclear factor of activated T cells; STAT6, signal transducer and activator of transcription 6; GATA3, GATA binding protein 3; Th2, T helper 2; IgE, immunoglobulin E; AHR, airway hyperresponsiveness.

**Figure 2 ijms-24-11970-f002:**
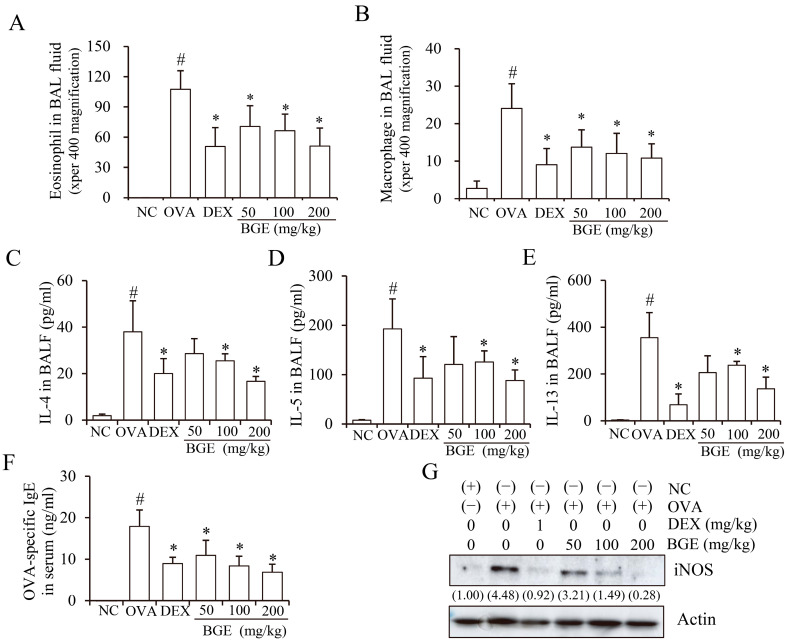
Effects of black ginseng extract (BGE) on allergic airway inflammatory responses in ovalbumin (OVA)-sensitized mice with allergic airway inflammation. (**A**,**B**) The number of immune cells (eosinophils/macrophages) in bronchoalveolar lavage fluid (BALF) was determined using cell counting. (**C**–**F**) The secretion levels of interleukin-4 (**C**), -5 (**D**), and -13 (**E**) in bronchoalveolar lavage fluid, and the production of serum immunoglobulin E (**F**) in OVA-sensitized mice with allergic airway inflammation, as determined using an enzyme-linked immunosorbent assay. (**G**) The expression of inducible nitric oxide synthase (iNOS) in lung tissue lysates, detected using Western blot analysis with antibodies against iNOS and β-actin as a loading control. The numbers underneath the bands indicate the relative band intensity (fold change relative to control). Dexamethasone (DEX) was used as a positive control. NC: normal control; OVA: ovalbumin-sensitized; DEX: 1 mg/kg DEX-treated OVA mice; BGE 50: 50 mg/kg BGE-treated OVA mice; BGE 100: 100 mg/kg BGE-treated OVA mice; and BGE 200: 200 mg/kg BGE-treated OVA mice. The values are expressed as mean ± standard deviation (*n* = 6). # *p* < 0.01: significantly different compared with the NC; * *p* < 0.05: significantly different compared with the OVA-induced group.

**Figure 3 ijms-24-11970-f003:**
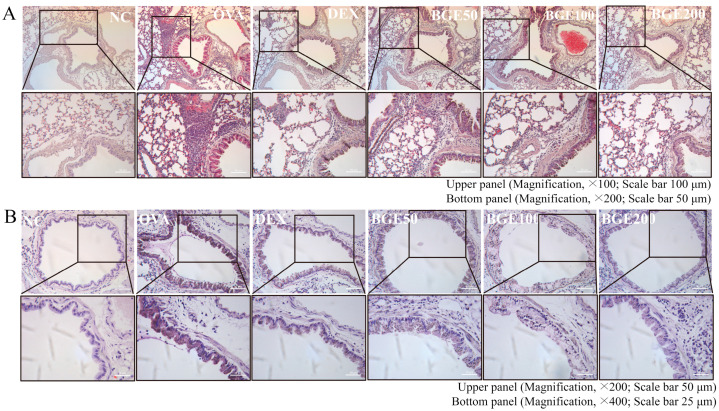
Effects of black ginseng extract (BGE) on inflammatory cell recruitment and mucus secretion in the airway of ovalbumin (OVA)-sensitized asthma mice. (**A**) Hematoxylin and eosin staining was performed to visualize the accumulation of inflammatory cells in the peribronchial region. Black arrowheads indicate the regions where inflammatory cells accumulated. Magnification, ×100; scale bar, 100 μm. Images obtained by magnifying the corresponding portion of the black rectangle in the upper panel are shown in the bottom panel. Magnification, ×200; scale bar, 50 μm. (**B**) Mucus in the airways was stained using a periodic acid–Schiff stain. Blue arrowheads indicate mucus accumulation. Magnification, ×200; scale bar, 50 μm. Images obtained by magnifying the corresponding portion of the black rectangle in the upper panel are shown in the bottom panel. Magnification, ×400; scale bar, 25 μm. Dexamethasone (DEX) was used as a positive control. NC: normal control; OVA: ovalbumin-treated; DEX: 1 mg/kg DEX-treated OVA mice; BGE 50: 50 mg/kg BGE-treated OVA mice; BGE 100: 100 mg/kg BGE-treated OVA mice; and BGE 200: 200 mg/kg BGE-treated OVA mice group. Representative images were randomly selected from three to four images.

**Figure 4 ijms-24-11970-f004:**
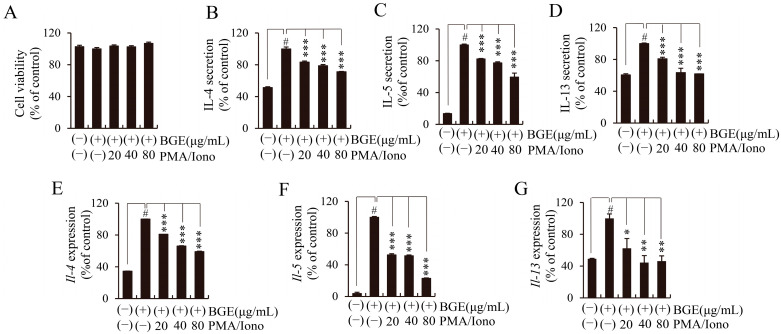
Effects of black ginseng extract (BGE) on the expression of T helper 2 cytokines in phorbol 12-myristate 13-acetate plus ionomycin (PMA/Iono)-stimulated EL4 cells. (**A**) Cell viability was measured using the cell counting kit-8 assay. (**B**–**D**) The secretion levels of interleukin (IL)-4, -5, and -13 were assessed using enzyme-linked immunosorbent assay kits. (**E**–**G**) The mRNA expression levels of *Il-4*, *-5*, and *-13* were evaluated using quantitative real-time polymerase chain reaction. Bar graphs represent the means ± standard deviation in triplicate in one representative experiment from a total of three independent experiments. # *p* < 0.001 vs. negative control group (without PMA/Iono); * *p* < 0.05; ** *p* < 0.01, and *** *p* < 0.001 for comparison with controls, which were treated with PMA/Iono alone.

**Figure 5 ijms-24-11970-f005:**
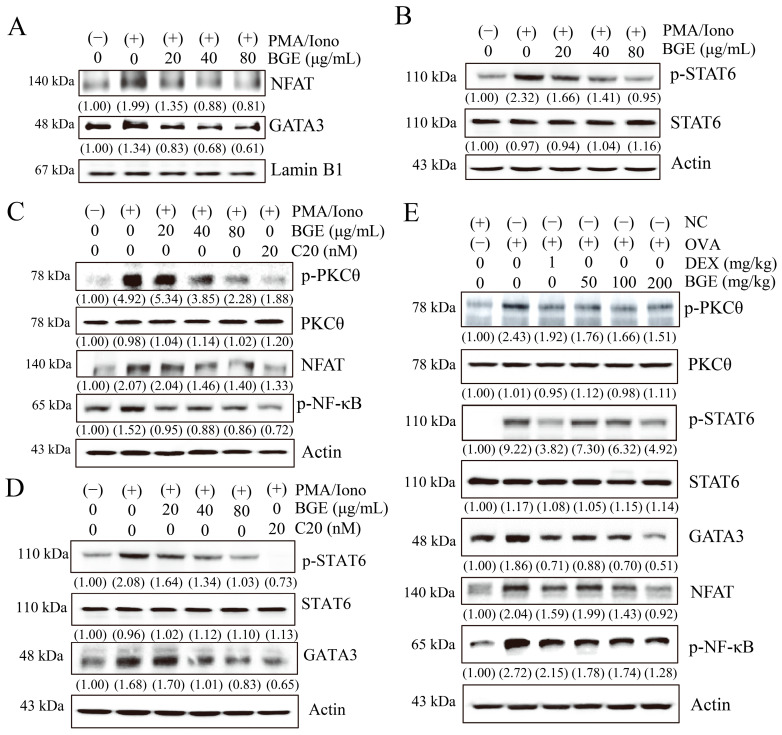
Effects of black ginseng extract (BGE) on signal transduction in T helper 2 immune responses. (**A**,**B**) The nuclear fraction (panel (**A**)) and total cell lysates (panel (**B**)) of EL4 cells were assayed using Western blot analysis with antibodies against nuclear factor of activated T cells (NFAT), GATA binding protein 3 (GATA3), phosphorylated signal transducer and activator of transcription 6 (p-STAT6), or STAT6. Lamin B1 and β-actin were used as loading controls for the nuclear fraction and total cell lysates, respectively. (**C**,**D**) Total cell lysates were estimated using Western blot analysis with antibodies against phosphorylated protein kinase C theta (p-PKCθ), PKCθ, NFAT, phosphorylated nuclear factor kappa B (p-NF-κB), p-STAT6, STAT6, and GATA3. C20 was used as a specific inhibitor of PKCθ. Panels (**C**,**D**) show the results of cells that were stimulated with PMA/Iono for 30 min and 24 h, respectively. β-actin was used as a loading control. (**E**) Western blot analysis using the lung tissue lysates of ovalbumin-induced mice with allergic airway inflammation with antibodies against p-PKCθ, PKCθ, p-STAT6, STAT6, GATA3, NFAT, and p-NF-κB, and β-actin as a loading control. The blots are representative of two independent experiments. The numbers underneath the bands indicate the relative band intensity (fold change relative to control). NC: normal control; OVA: ovalbumin-treated; DEX: 1 mg/kg DEX-treated OVA mice; BGE 50: 50 mg/kg BGE-treated OVA mice; BGE 100: 100 mg/kg BGE-treated OVA mice; and BGE 200: 200 mg/kg BGE-treated OVA mice.

**Figure 6 ijms-24-11970-f006:**
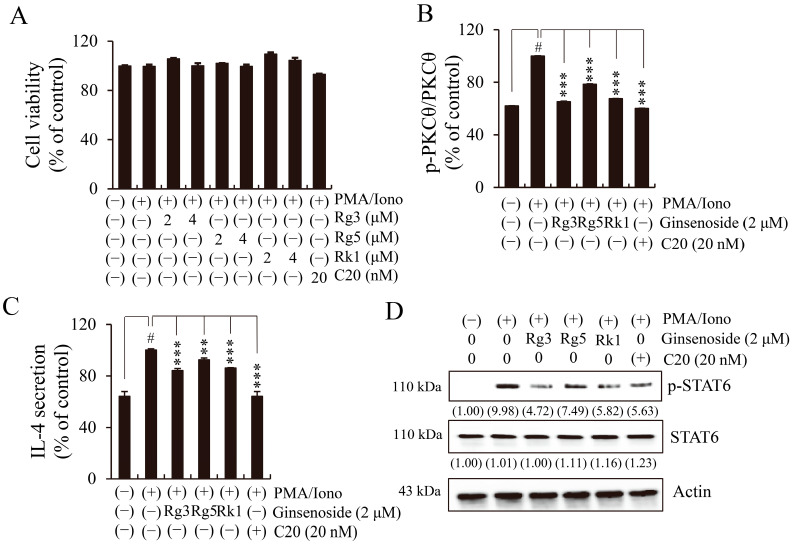
Effects of three ginsenosides (Rg3, Rg5, and Rk1) on protein kinase C theta (PKCθ) phosphorylation and the interleukin (IL)-4/signal transducer and activator of transcription (STAT) 6 pathway in phorbol 12-myristate 13-acetate plus ionomycin (PMA/Iono)-stimulated EL4 cells. C20 was used as a specific inhibitor of PKCθ. (**A**) Cell viability was measured using the cell counting kit-8 method. (**B**) The phosphorylation of PKCθ, as measured using a commercialized PKCθ (phospho Thr538) cell-based enzyme-linked immunosorbent assay (ELISA) kit. (**C**) The levels of IL-4 secretion, as measured using an ELISA kit. Bar graphs indicate the means ± standard deviation in triplicate in one representative experiment from a total of three independent experiments. # *p* < 0.001 vs. negative control group (without PMA/Iono); ** *p* < 0.01, and *** *p* < 0.001 for comparison with controls treated with PMA/Iono alone. (**D**) The total cell lysates were analyzed using Western blotting with antibodies against phosphorylated signal transducer and activator of transcription 6 (p-STAT6) and STAT6. β-actin was used as a loading control for total lysates. The numbers underneath the bands indicate the relative band intensity (fold change relative to the control). The blots are representative of two independent experiments.

## Data Availability

The data presented in this study are all contained within the article.

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
