# Peer review of "Black Ginseng Extract Exerts Potentially Anti-Asthmatic Activity by Inhibiting the Protein Kinase Cθ-Mediated IL-4/STAT6 Signaling Pathway"

_ijms, 2023, doi:10.3390/ijms241511970_

Round 1

Reviewer 1 Report

The manuscript by Yu Na Song et al. seems promising, but needs a lot of revision.

My comments:

1. The layout of the manuscript does not comply with the IJMS requirements - please correct it

2. figure 1 appears in the text much later than the first reference to this figure - please correct

3. Figure 1 should not be blocked - it should be divided into at least two figures - even if it is necessary to present the chromatograms separately (in a separate figure)

4. The methodology is very poorly described - it is not enough to specify which company the tests were used, please provide their catalog numbers exactly (which uniquely identify the test kit)

5. I have doubts about the attached photos of the original gels/blots - these are still only their fragments, not whole gels - please attach the whole, original and unprocessed photos of gels/blots

6. Figure 6 is indicated only in the conclusions. The "conclusions" section is not the right place to present figures. I would suggest moving this figure to the introduction and formulating the aim of the research in such a way that it refers to it.

7. Currently, the conclusions describe the results - in this form, these are not conclusions, but a brief description of the results - please draw conclusions from this description and provide these conclusions here

8. Athors do not indicate the weaknesses and limitations of their experience - I am asking for a critical approach to the results of their experiments and pointing out their limitations and weaknesses

Author Response

Point by Point Responses

The followings are the comments of reviewers, represented by a bolded Q). Our responses are indicated in blue.

 All revised parts of our manuscript can be found in the blue-colored parts of the main text or through the “change tracking”. We sincerely hope our response satisfies the reviewer’s request.

Reviewer #1 (Remarks to the Author):

Reviewer’s major comment:

Q1-1) The layout of the manuscript does not comply with the IJMS requirements - please correct it.

===>Thanks for pointing out the incompleteness. As you requested, we have adjusted the order of the sections to Introduction, Results, Discussion, Materials and methods, and Conclusions to comply with IJMS requirements, and relevant references, tables, and figure numbers been updated accordingly.

Q1-2) figure 1 appears in the text much later than the first reference to this figure - please correct

===> To address the reviewer's concerns, we modified Figure 1 by moving from Figures 1A and 1B to Supplementary Figures S1A and S1B, so that the first reference was properly designated.

Q1-3) Figure 1 should not be blocked - it should be divided into at least two figures - even if it is necessary to present the chromatograms separately (in a separate figure)

===>According to the reviewer’s comments, we have separated HPLC profiling in Figure 1 by moving from Figures 1A and 1B to Supplementary Figures S1A and S1B. Therefore, we have newly added the relevant Supplementary Figures S1 and revised the manuscript (results p.3, lines 115-117; Materials and methods p.10, line 375, line 385; Supplementary materials p.13, lines 524-525).

Q1-4) The methodology is very poorly described - it is not enough to specify which company the tests were used, please provide their catalog numbers exactly (which uniquely identify the test kit)

===> Thanks for pointing out the insufficient. we added a “Materials and methods” section by accurately providing a catalog number that uniquely identifies the test kit (Materials and methods p. 11, line 390, 412, 413, 415, 420, 426, 427, 431, 450, 451, 458, 464, 471, 473, 475, 487).

Q1-5) I have doubts about the attached photos of the original gels/blots - these are still only their fragments, not whole gels - please attach the whole, original and unprocessed photos of gels/blots

==>This point is well taken. To address the reviewer's concerns, we have newly modified Supplementary Figures S2, S3 and S4 by adding uncropped and unprocessed blots along with protein size markers. Although, there are membrane cuts to simultaneously identify multiple target proteins of different sizes on one membrane, they are identified by protein size markers. I hope you understand these points. The revised manuscript was indicated in Supplementary materials p.13, lines 525-526.

Q1-6) Figure 6 is indicated only in the conclusions. The "conclusions" section is not the right place to present figures. I would suggest moving this figure to the introduction and formulating the aim of the research in such a way that it refers to it.

===>We absolutely agree with the reviewer's comment. As suggested by the reviewer, Figure 6 was moved to the introduction and the order was changed to Figure 1, and the purpose of the study was formulated with reference to it in revised manuscript (Introduction in p. 3, lines 101-102).

Q1-7) Currently, the conclusions describe the results - in this form, these are not conclusions, but a brief description of the results - please draw conclusions from this description and provide these conclusions here.

===> Thank you for reviewer’s valuable comments. According to the reviewer’s suggestion, the conclusion (p. 13, lines 512-522) was rewritten as follows:

“Here, we demonstrate that BGE can alleviate allergic airway inflammation in OVA-induced asthmatic mice and inactivate PKCθ and the inflammatory transcription factors associated with Th2 cytokine expression. The key findings of this study are that BGE inhibits the PKCθ-mediated signaling molecules related to a central pathway in allergic airway asthma and the corticosteroid drug DEX, used as a positive control, has similar efficacy and mechanisms to BGE. Considering the side effects of current DEX therapies, BGE could be a potential adjunct to a safer and more effective treatment for allergic airway inflammation. Thus, we propose BGE as a promising therapeutic, pharmacological, and nutraceutical candidate for allergic asthma management. An important future direction includes validating our findings in human studies with the overall aim of evaluating the therapeutic potential of BGE in allergic asthma.”

Q1-8) Authors do not indicate the weaknesses and limitations of their experience - I am asking for a critical approach to the results of their experiments and pointing out their limitations and weaknesses.

===> Thank you for your constructive comments. For this reason, we propose the following paragraph at the end of the discussion section (p. 10, lines 348-356):

“Like most studies, the present study has some limitations. In this study, we only used EL4 cells, which are a mouse T cell lymphoma cell line, and additional cell lines such as primary human cells were not used. Although EL4 cells are widely used as a model system for studying the expression of T cell markers and molecular mechanisms by activating the T cell receptor using substances such as PMA and Iono [50, 51], primary human cells from patients with asthma will provide more valuable supporting evidence for the anti-inflammatory effect of BGE. Despite these limitations, our study provides significant contribution to the existing literature. However, future studies should be designed to closely relate to human application tests by using primary human cells from patients with asthma.”

Reviewer 2 Report

The manuscript “Black ginseng extract exerts anti-asthmatic activity by inhibiting the protein kinase Cθ-mediated IL-4/STAT6 signaling pathway” is a very carefully written investigation into the effects of black ginseng and some components of this preparation on the inflammatory profile in asthma, defined as Th2 cytokines in BALF, IgE in serum and iNOS in lung tissue lysates. The models used were OVA induced mice and PMA and ionomycin stimulated EL4 lymphoblasts. The effect of black ginseng on these parameters was investigated in mice after oral administration for five days simultaneously with OVA. In the cells, production of cytokines IL-4, IL-5 and IL-13 was investigated with or without black ginseng. In summary, black ginseng inhibited phosphorylation of STAT6 in a PKC-dependent manner, resulting in decreased inflammatory responses.

The manuscript is well structured and the experiments comprehensively described. The statistical analysis is correctly done and the references are carefully chosen. The results are supported by the experimental data, but there are some concerns which need to be addressed:

Major concerns:

1) Experimental animals: The manuscript states that Six-week-old female BALB/c mice were used for the experiments. This poses some problems:

A) Only one sex

B) Very young mice

C) Was there no acclimatization period?

Please justify these points and comment on how the results can be generalized.

2) Figure 2: The resolution and magnification is too low for visualization of A) inflammatory cells and B) goblet cells. The arrows point to something, but it is impossible to see what it is. Please magnify important features in insets.

3) Line 480 of the manuscript: Taken together, we propose BGE as a promising theraeutic, pharmacological, or nutraceutical candidate for asthma management.

What happens when you combine it with your standard asthma treatment?

Minor concerns:

1) Line 115: And the sum of ginsenosides Rg3, Rk1, and Rg5 was 7.36 mg/g, which is suitable for BGE's quality control criteria, and a representative HPLC chromatogram of BGE is shown in Figure 1A and 1B.

A sentence cannot start with “And”. Please rewrite.

2) Line 275: However, BGE and DEX treatment indicated better histopathological changes

Please explain what you mean by “better histopathological changes”. Do you mean more normal tissue?

The language is generally quite good, with some exceptions, please see the comments to authors. The text would benefit from language editing.

Author Response

Point by Point Responses

The followings are the comments of reviewers, represented by a bolded Q). Our responses are indicated in blue.

 All revised parts of our manuscript can be found in the blue-colored parts of the main text or through the “change tracking”. We sincerely hope our response satisfies the reviewer’s request.

Reviewer #2 (Remarks to the Author):

The manuscript “Black ginseng extract exerts anti-asthmatic activity by inhibiting the protein kinase Cθ-mediated IL-4/STAT6 signaling pathway” is a very carefully written investigation into the effects of black ginseng and some components of this preparation on the inflammatory profile in asthma, defined as Th2 cytokines in BALF, IgE in serum and iNOS in lung tissue lysates. The models used were OVA induced mice and PMA and ionomycin stimulated EL4 lymphoblasts. The effect of black ginseng on these parameters was investigated in mice after oral administration for five days simultaneously with OVA. In the cells, production of cytokines IL-4, IL-5 and IL-13 was investigated with or without black ginseng. In summary, black ginseng inhibited phosphorylation of STAT6 in a PKC-dependent manner, resulting in decreased inflammatory responses.

The manuscript is well structured and the experiments comprehensively described. The statistical analysis is correctly done and the references are carefully chosen. The results are supported by the experimental data, but there are some concerns which need to be addressed:

Reviewer’s major comment:

Q2-1) Experimental animals: The manuscript states that Six-week-old female BALB/c mice were used for the experiments. This poses some problems.

  1. A) Only one sex

===>In our asthma experiment, female was preferentially selected over males because women are known to have a higher prevalence of asthma than men [1]. Indeed, many in vivo studies of allergic asthma treatment have selected females as shown in the table below.

However, the need to investigate gender-related effects of asthma treatment is highlighted in order to plan future human trial protocols [1]. In later experiments, we will study using both males and females to expand the study.

Table. sex selection and starting weeks in previous studies

experimental design

NO.

Sex selection

Starting weeks

References

1

female

Six-week

[2]

2

female

Six-week

[3]

3

female

Six-week

[4]

4

female

7–8 weeks

[5]

5

male

8 weeks

[6]

6

female

6–8 weeks

[7]

7

female

4–6 weeks old; weighing 18–22 g

[8]

female

Six-week (weighing approximately 18 g)

   In this study

  1. B) Very young mice

===>Thank you for valuable comments. We used female and 6 weeks (weighing approximately 18 g) in an asthmatic mouse model based on the following references. Therefore, female and 6 weeks used in this study is considered to be within a reasonable selection.

References

  1. Fuseini, H.; Newcomb, D. C., Mechanisms Driving Gender Differences in Asthma. Curr Allergy Asthma Rep 2017, 17, (3), 19.
  2. Shin, N. R.; Lee, A. Y.; Park, G.; Ko, J. W.; Kim, J. C.; Shin, I. S.; Kim, J. S., Therapeutic Effect of Dipsacus asperoides C. Y. Cheng et T. M. Ai in Ovalbumin-Induced Murine Model of Asthma. Int J Mol Sci 2019, 20, (8).
  3. Kim, M. G.; Kim, S. M.; Min, J. H.; Kwon, O. K.; Park, M. H.; Park, J. W.; Ahn, H. I.; Hwang, J. Y.; Oh, S. R.; Lee, J. W.; Ahn, K. S., Anti-inflammatory effects of linalool on ovalbumin-induced pulmonary inflammation. Int Immunopharmacol 2019, 74, 105706.
  4. Park, J. W.; Choi, J.; Lee, J.; Park, J. M.; Kim, S. M.; Min, J. H.; Seo, D. Y.; Goo, S. H.; Kim, J. H.; Kwon, O. K.; Lee, K.; Ahn, K. S.; Oh, S. R.; Lee, J. W., Methyl P-Coumarate Ameliorates the Inflammatory Response in Activated-Airway Epithelial Cells and Mice with Allergic Asthma. Int J Mol Sci 2022, 23, (23).
  5. Hwang, Y. H.; Hong, S. G.; Mun, S. K.; Kim, S. J.; Lee, S. J.; Kim, J. J.; Kang, K. Y.; Yee, S. T., The Protective Effects of Astaxanthin on the OVA-Induced Asthma Mice Model. Molecules 2017, 22, (11).
  6. Lin, C. C.; Chuang, K. C.; Chen, S. W.; Chao, Y. H.; Yen, C. C.; Yang, S. H.; Chen, W.; Chang, K. H.; Chang, Y. K.; Chen, C. M., Lactoferrin Ameliorates Ovalbumin-Induced Asthma in Mice through Reducing Dendritic-Cell-Derived Th2 Cell Responses. Int J Mol Sci 2022, 23, (22).
  7. Bai, D.; Sun, T.; Lu, F.; Shen, Y.; Zhang, Y.; Zhang, B.; Yu, G.; Li, H.; Hao, J., Eupatilin Suppresses OVA-Induced Asthma by Inhibiting NF-kappaB and MAPK and Activating Nrf2 Signaling Pathways in Mice. Int J Mol Sci 2022, 23, (3).
  8. Yang, Z.; Li, X.; Fu, R.; Hu, M.; Wei, Y.; Hu, X.; Tan, W.; Tong, X.; Huang, F., Therapeutic Effect of Renifolin F on Airway Allergy in an Ovalbumin-Induced Asthma Mouse Model In Vivo. Molecules 2022, 27, (12).

  1. C) Was there no acclimatization period?

===>We apologize for our mistake that not describe as below, and included it in revised manuscript (p.11, lines 399-400) as follows: “Six-week-old female BALB/c mice (weighing approximately 18 g) were purchased from Koatech Co. (Pyeongtaek, Republic of Korea) and acclimated to the environment for 1 week before the experiment.”

Q2-2) Figure 2: The resolution and magnification is too low for visualization of A) inflammatory cells and B) goblet cells. The arrows point to something, but it is impossible to see what it is. Please magnify important features in insets.

===> Thank you for your valuable comments. We revised it as much as possible to meet the reviewer’ suggestion. Please check Figures 3A and 3B in the revised manuscript (figure legends in p.5, lines 175-176, lines 178-180).

Q2-3) Line 480 of the manuscript: Taken together, we propose BGE as a promising therapeutic, pharmacological, or nutraceutical candidate for asthma management. What happens when you combine it with your standard asthma treatment?

===> In this study, we used the positive control dexamethasone (DEX). This drug, a corticosteroid drug that alleviates inflammation, is prescribed for the treatment of rheumatoid arthritis, allergies, and acute respiratory distress syndrome. Recently, DEX was mainly used also as a therapeutic drug of COVID-19 [1]. However, DEX remedy for high-dose and long periods can lead to various side effect including eye problems such as cataracts or glaucoma [2]. Importantly, we found that BGE and DEX show a similar inhibitory effect and molecular mechanism on allergic airway inflammatory responses in ovalbumin (OVA)-sensitized asthma mice. Taken into account the adverse side effects of current DEX therapeutic agents, we propose BGE as an adjuvant in combination with DEX to reduce the side effects and increase the inhibitory efficiency of allergenic airway inflammation.

References

  1. Noreen, S.; Maqbool, I.; Madni, A., Dexamethasone: Therapeutic potential, risks, and future projection during COVID-19 pandemic. Eur J Pharmacol 2021, 894, 173854.
  2. Abtahi, S. H.; Nouri, H.; Moradian, S.; Yazdani, S.; Ahmadieh, H., Eye Disorders in the Post-COVID Era. J Ophthalmic Vis Res 2021, 16, (4), 527-530.

Reviewer’s minor comment:

Q2-1) Line 115: And the sum of ginsenosides Rg3, Rk1, and Rg5 was 7.36 mg/g, which is suitable for BGE's quality control criteria, and a representative HPLC chromatogram of BGE is shown in Figure 1A and 1B. A sentence cannot start with “And”. Please rewrite.

===> Based on the reviewer's comment, we removed the word “And” . These revised sentence described in the revised manuscript (p.10, lines 373-375) as follow:

"The sum of ginsenosides Rg3, Rk1, and Rg5 was ~~~~~~~~~~~~~~~ shown in Figure S1."

Q2-2) Line 275: However, BGE and DEX treatment indicated better histopathological changes. Please explain what you mean by “better histopathological changes”. Do you mean more normal tissue?

===> yes, to clarify the meaning of the results, this revised sentence described in the revised manuscript (p.5, lines 163-164) as follow:

“However, BGE and DEX treatment alleviated these histopathological changes (Figure 3A).”

Reviewer 3 Report

With interest, I read the manuscript ijms-2520355. It is important to investigate the role and possible therapeutic applications of natural compounds (e.g. PMID: 35955959, PMID: 36983066).

Comments (no special order):

1.     The Authors model here allergic airway inflammation mimicking human atopic asthma. In most of the cases, it is well reported in this draft. However, sometimes one can read about asthmatic mice, mice with asthma, etc. These should be corrected. Also in the title, the Authors write about anti-asthmatic effects (better -> potentially anti-asthmatic effects, etc.).

2.     The basic (and in fact only reliable) proof that the model worked is BAL cell count, especially/specifically BAL eosinophils. Maybe I overlooked something but I cannot see those data. Please, provide for all animal groups. This comment is critical.

3.     Lines 47-56. Between steroids and natural drugs, please, add biologicals (PMID: 33926084).

4.     EL4 cell line. Why particularly this cell line? Why not human cells, especially primary cells? If not yet done, please, add to the limitations of the study.

5.     Graphs in several figures are too small too read. Please, amend.

6.     Please, verify and correct the name of murine genes, e.g. in Figure 3.

7.     The Authors focus on PKCθ. Also PKCζ has been reported to be crucial for the biology of T cells and the development of allergic disorders and it could be affected by some natural compounds through the epigenetic mechanisms (PMID: 28159873, 33668787, 34063174, 34884454). Please, address in the Discussion.

8.      Again, maybe I overlooked something but I cannot see anything about the ethical approval. This must be provided. This comment is critical.

Minor amendments

Author Response

Point by Point Responses

The followings are the comments of reviewers, represented by a bolded Q). Our responses are indicated in blue.

 All revised parts of our manuscript can be found in the blue-colored parts of the main text or through the “change tracking”. We sincerely hope our response satisfies the reviewer’s request.

Reviewer #3 (Remarks to the Author):

With interest, I read the manuscript ijms-2520355. It is important to investigate the role and possible therapeutic applications of natural compounds (e.g. PMID: 35955959, PMID: 36983066).

Q3-1) The Authors model here allergic airway inflammation mimicking human atopic asthma. In most of the cases, it is well reported in this draft. However, sometimes one can read about asthmatic mice, mice with asthma, etc. These should be corrected. Also in the title, the Authors write about anti-asthmatic effects (better -> potentially anti-asthmatic effects, etc.).

===> Thank you for your constructive comments. In accordance with the reviewer's advice, we modified the wording in the manuscript from "asthmatic mice" to "allergic asthmatic mice" and "mice with asthma" to "mice with allergic asthma". In addition, we added “potentially” in the title and in the revised manuscript (title in p.1 line 2; abstract in p.1 line 31, 36, 38, 39; introduction p.3 line 103; results in p.3 line 128, 135, 238, 240; discussion in p.9 line 295, 301).

Q3-2) The basic (and in fact only reliable) proof that the model worked is BAL cell count, especially/specifically BAL eosinophils. Maybe I overlooked something but I cannot see those data. Please, provide for all animal groups. This comment is critical.

===> We absolutely agree with the reviewer’s comments. To address the reviewer’s concerns, we newly added Figure 2A by measuring inflammatory cell count (eosinophil/macrophage) in the BALF from OVA-induced asthma model. In these result, the OVA-induced asthma model exhibited a marked increase in the number of eosinophils and macrophages, compared with the normal control (NC) mice (Figure 2AB, second bar, respectively). However, BGE and dexamethasone (DEX) administration significantly reduced the number of eosinophils and macrophages compared to the OVA-induced asthma model. BGE was comparable to that of a 1 mg/kg administration of DEX, which was used as a positive control (Figure 2A and 2B). These new data added in Figure 2A and 2B in the revised manuscript (results in p.3, line 114, lines 118-126, line 136; figure legends in p.4, line 141; lines 142-143; Materials and methods p.11, lines 417-423).

Q3-3) Lines 47-56. Between steroids and natural drugs, please, add biologicals (PMID: 33926084). 

===> Thank you for your valuable comments. We added the word "biologicals" and cited references (PMID: 33926084) in revised manuscript (Introduction in p.2, lines 54-55)

Q3-4) EL4 cell line. Why particularly this cell line? Why not human cells, especially primary cells? If not yet done, please, add to the limitations of the study.

===> Thank you for your constructive comments. For this reason, we propose the following paragraph at the end of the discussion section (p.10, lines 348-356):  

“Like most studies, the present study has some limitations. In this study, we only used EL4 cells, which are a mouse T cell lymphoma cell line, and additional cell lines such as primary human cells were not used. Although EL4 cells are widely used as a model system for studying the expression of T cell markers and molecular mechanisms by activating the T cell receptor using substances such as PMA and Iono [50, 51], primary human cells from patients with asthma will provide more valuable supporting evidence for the anti-inflammatory effect of BGE. Despite these limitations, our study provides significant contribution to the existing literature. However, future studies should be designed to closely relate to human application tests by using primary human cells from patients with asthma.”

Q3-5) Graphs in several figures are too small to read. Please, amend.

===>We apologize for any inconvenience. According to the reviewer’s suggestion, we revised greatly enlarged the text in the Figure 2, Figure 3, Figure 4, Figure 5, and Figure 6 in revised manuscript.

Q3-6) Please, verify and correct the name of murine genes, e.g. in Figure 3.

===> Based on the reviewer's suggestion, IL-4, -5, and -13 have been corrected to the mouse gene names Il-4, Il-5, and Il-13 in Figure 4 (on p.6) and revised manuscript (results in p.6, lines 195, 197; figure legend in p.6 line 206; Materials and methods p.12 line 477,478, 480, 481).

Q3-7) The Authors focus on PKCθ. Also PKCζ has been reported to be crucial for the biology of T cells and the development of allergic disorders and it could be affected by some natural compounds through the epigenetic mechanisms (PMID: 28159873, 33668787, 34063174, 34884454). Please, address in the Discussion.

===> Thank you for your constructive comments. We added in “Discussion” section (p.9, lines 319-323) and cited references (PMID: 28159873, 33668787, 34063174, 34884454) as follows:

“Of PKC isozymes, PKCζ has been reported to be crucial for the biology of T cells and the development of allergic disorders. Some natural compounds can affect PKCζ through epigenetic mechanisms [40-43]. Moreover, inhibition of PKCδ activation was found to reduce asthma attacks in a murine model [44]. PKCθ activity in particular is important for Th2-mediated responses in allergic lung inflammation [45].”

Q3-8) Again, maybe I overlooked something but I cannot see anything about the ethical approval. This must be provided. This comment is critical.

===> Thanks for pointing out the incomplete. As you requested, we added Ethics statement in the “Material and methods” section (on p.13, lines 498-503).

Round 2

Reviewer 1 Report

Thank you for the changes made.

Author Response

We thank  the reviewer for your thoughtful suggestions and insights.

Reviewer 3 Report

I am very grateful to the Authors for addressing my comments rather well! Thank you!

Only in the case of my comment 1 (see below, along with the answer), there have been misunderstanding. Possibly, I am guilty here as I was not clear enough.

My previous comment 1:

Q3-1) The Authors model here allergic airway inflammation mimicking human atopic asthma. In most of the cases, it is well reported in this draft. However, sometimes one can read about asthmatic mice, mice with asthma, etc. These should be corrected. Also in the title, the Authors write about anti-asthmatic effects (better -> potentially anti-asthmatic effects, etc.).

Your answer to my previous comment 1:

===> Thank you for your constructive comments. In accordance with the reviewer's advice, we modified the wording in the manuscript from "asthmatic mice" to "allergic asthmatic mice" and "mice with asthma" to "mice with allergic asthma". In addition, we added “potentially” in the title and in the revised manuscript (title in p.1 line 2; abstract in p.1 line 31, 36, 38, 39; introduction p.3 line 103; results in p.3 line 128, 135, 238, 240; discussion in p.9 line 295, 301).

My additional comment to this point:

Since you model here allergic airway inflammation mimicking human atopic asthma, it should never be written “allergic asthmatic mice”, “mice with allergic asthma”, “asthmatic mice”, “mice with asthma”, etc., but “mice with allergic airway inflammation". I am sorry that I was not clear enough in my first comment but I kindly ask you to change it throughout the manuscript. Besides, I have no further comments and congratulate you on this work!

Only very minor amendments required.

Author Response

Point by Point Responses

The followings are the comments of reviewers, represented by a bolded Q). Our responses are indicated in blue.

 All revised parts of our manuscript can be found in the blue-colored parts of the main text or through the “change tracking”. We sincerely hope our response satisfies the reviewer’s request.

Reviewer #3 (Comments and Suggestions for Authors)

I am very grateful to the Authors for addressing my comments rather well! Thank you! Only in the case of my comment 1 (see below, along with the answer), there have been misunderstanding. Possibly, I am guilty here as I was not clear enough.

 My additional comment to this point:

Q3-1) Since you model here allergic airway inflammation mimicking human atopic asthma, it should never be written “allergic asthmatic mice”, “mice with allergic asthma”, “asthmatic mice”, “mice with asthma”, etc., but “mice with allergic airway inflammation". I am sorry that I was not clear enough in my first comment but I kindly ask you to change it throughout the manuscript. Besides, I have no further comments and congratulate you on this work!

===>Thank you for your constructive comments. In accordance with the reviewer's advice, we changed the wording in the manuscript from "allergic asthmatic mice", "mice with allergic asthma", and "asthmatic mice" to "mice with allergic airway inflammation ". We added in the revised manuscript (abstract in p.1 line 25, 32, 38; introduction in p.2 line 73, 104, 110; results in p.3 line 116, 121, 128, 132, 135, 146, 149, 160, 166, 174, 245, 258; discussion in p.9 line 306, 307, 322, 350; conclusions p.13 line 520).
